# Levofloxacin and Ciprofloxacin Co-Crystals with Flavonoids: Solid-State Investigation for a Multitarget Strategy against *Helicobacter pylori*

**DOI:** 10.3390/pharmaceutics16020203

**Published:** 2024-01-30

**Authors:** Cecilia Fiore, Federico Antoniciello, Davide Roncarati, Vincenzo Scarlato, Fabrizia Grepioni, Dario Braga

**Affiliations:** 1Department of Chemistry “Giacomo Ciamician”, University of Bologna, Via Selmi 2, 40126 Bologna, Italy; fabrizia.grepioni@unibo.it (F.G.); dario.braga@unibo.it (D.B.); 2Department of Applied Science and Technology (DISAT), Politecnico di Torino, Corso Duca degli Abruzzi 24, 10129 Torino, Italy; 3Department of Pharmacy and Biotechnology (FaBiT), University of Bologna, Via Selmi 3, 40126 Bologna, Italy; davide.roncarati@unibo.it (D.R.); vincenzo.scarlato@unibo.it (V.S.)

**Keywords:** co-crystals, crystal engineering, flavonoids, antimicrobials, HP1043, essential transcription factor, *Helicobacter pylori*

## Abstract

In this paper, we address the problem of antimicrobial resistance in the case of *Helicobacter pylori* with a crystal engineering approach. Two antibiotics of the fluoroquinolone class, namely, levofloxacin (LEV) and ciprofloxacin (CIP), have been co-crystallized with the flavonoids quercetin (QUE), myricetin (MYR), and hesperetin (HES), resulting in the formation of four co-crystals, namely, LEV∙QUE, LEV∙MYR, LEV_2_∙HES, and CIP∙QUE. The co-crystals were obtained from solution, slurry, or mechanochemical mixing of the reactants. LEV∙QUE and LEV∙MYR were initially obtained as the ethanol solvates LEV∙QUE∙xEtOH and LEV∙MYR∙xEtOH, respectively, which upon thermal treatment yielded the unsolvated forms. All co-crystals were characterized by powder X-ray diffraction and thermal gravimetric analysis. The antibacterial performance of the four co-crystals LEV∙QUE, LEV∙MYR, LEV_2_∙HES, and CIP∙QUE in comparison with that of the physical mixtures of the separate components was tested via evaluation of the minimal inhibitory concentration (MIC) and minimal bactericidal concentration (MBC). The results obtained indicate that the association with the co-formers, whether co-crystallized or forming a physical mixture with the active pharmaceutical ingredients (API), enhances the antimicrobial activity of the fluoroquinolones, allowing them to significantly reduce the amount of API otherwise required to display the same activity against *H. pylori*.

## 1. Introduction

*Helicobacter pylori* is a Gram-negative and transmissible pathogen that colonizes the human stomach, causing chronic infections that result in several gastric disorders, such as peptic ulceration, gastric adenocarcinoma, and MALT lymphoma. Most gastric cancer diagnoses are indeed attributable to *H. pylori* infection [1,2]. Like other superbugs, *H. pylori* rapidly develops resistance to the standard therapies, leading to a significant decrease in their efficacy and eradication cure rates between 70 and 50% [3]. Moreover, only a few antibiotics (such as amoxicillin, clarithromycin, metronidazole, tetracycline, levofloxacin, and rifabutin) can be used effectively for the eradication of *H. pylori* in clinical practice, typically when administered to patients in combined therapies with a proton-pump inhibitor (PPI) and/or a bismuth component that brings additional antibiotic effect and mucosal protection against aggressive factors [4,5]. Following the limited choice of effective therapeutics and the extensive use of certain antibiotics in the general population, rapid development of primary antibiotic resistance has been noted in *H. pylori* [3]. 

The fluoroquinolones levofloxacin and ciprofloxacin show a broad spectrum of activity against Gram-positive and Gram-negative bacteria. Levofloxacin-based regimens to treat *H. pylori* infections are usually triple-therapies including a PPI and amoxicillin [6,7,8,9]. It has been observed in recent studies that the efficacy of levofloxacin-containing therapy is decreasing, most likely due to increased primary resistance [3,10]. In order to offer an alternative to the standard therapies for the treatment of *H. pylori* infection, in a few studies, the efficacy of a ciprofloxacin-based regimen has also been explored [11,12].

The World Health Organization (WHO) [13] and the World Gastroenterology Organization (WGO) [14] have included *H. pylori* in their list of “priority pathogens” for which new antibiotics are urgently needed, and with this pressing need for novel therapeutic options, the scientific community’s interest in traditional medicine and the use of natural products as sources of novel antibacterial drugs has been reinforced [15]. 

It is well established that co-crystallization is a viable crystal engineering route to the synthesis of new materials and/or to the enhancement of the properties of active molecules [16,17]. In recent publications, we have shown that the solid-state association of active ingredients, such as ciprofloxacin and antibiotics of the cephalosporin class, with molecular components belonging to the GRAS (Generally Recognized as Safe) family allows us to enhance and/or alter the overall antimicrobial performance of the antibiotics [18,19]. 

Following the same approach, we now focus on bioactive compounds isolated from natural sources, which have gained interest in the scientific community due to their beneficial effects on human health [20,21]. Flavonoids are a large family of naturally occurring bioactive compounds present in various species and in a wide variety. Plant flavonoids play an important role in the protection against pathogenic microorganisms, such as bacteria, fungi, and viruses [15,22,23,24].

In order to tackle the antimicrobial resistance (AMR) problem in the case of *H. pylori*, we have explored the effect on the antibiotic activity of the co-crystallization of two antibiotics of the fluoroquinolone class, namely, levofloxacin and ciprofloxacin, with three natural compounds of the flavonoid family: quercetin, myricetin, and hesperetin. Quercetin (QUE) is an important phytochemical, a flavonol, belonging to the flavonoid group of polyphenols. It is widely distributed in various fruits, vegetables, and beverages as well as in flowers, leaves, and seeds [25]. QUE possesses many pharmacological activities such as antioxidant [26,27], anticancer [28,29], anti-inflammatory [30,31,32], antimicrobial [32,33,34], etc. Myricetin (MYR) is a flavonoid of the flavone type, present as well in many vegetables, fruits, nuts, berries, and herbs [35]; MYR exhibits antioxidant properties, free radical-scavenging effects, and more beneficial properties [36,37,38,39,40,41,42]. Hesperetin (HES) is another flavonoid, of the flavanone class [43]; HES possesses different activities as well, such as antioxidant, anti-inflammatory, antimicrobial, and anticarcinogenic [44,45,46].

The antibacterial activity and bioavailability of flavonoids are affected by various parameters, such as molecular conformation, hydrophobicity, solubility, etc. [47,48]. The exact mechanisms of the antibacterial effects of flavonoids remain unclear, although several mechanisms of action have been proposed, such as interference with bacterial DNA synthesis, bacterial movement, cytoplasmic membrane permeability, and the inhibition of bacterial metalloenzymes [22,33,49].

As reported for other phytochemicals, the antimicrobial activity of flavonoids appears multifactorial by acting against different molecular targets in the pathogen instead of having one specific action site [22,33]. 

Bacterial transcriptional and post-transcriptional regulators (TRs and PTRs) have emerged for their significant potential as novel drug targets [50]. Targeting a bacterial TR that controls a cluster of fundamental genes is an example of a specific multitargeting approach with high specificity for the pathogen. HP1043, also referred to as HsrA, is a conserved and essential TR that is involved in the tight regulation of a plethora of housekeeping and essential genes. This regulator is an orphan response regulator that lacks its cognate sensor histidine kinase. Due to its involvement in crucial pathways for the viability of the bacterium, HP1043 is a potentially optimal novel target for antimicrobial drug discovery [50,51,52]. Recent studies have demonstrated that a group of flavonoids can inhibit HP1043 DNA binding, resulting in antimicrobial activity against *H. pylori* [44,45]. Therefore, the concept of creating a co-crystal of an antibiotic with a flavonoid used as a co-former is one of the first examples of a multitarget approach that involves the essential bacterial TR HP1043.

In this work, we report the preparation and characterization of four co-crystals obtained by combining the fluoroquinolones levofloxacin (LEV) and ciprofloxacin (CIP) with the flavonoids quercetin (QUE), myricetin (MYR), and hesperetin (HES) (see Figure 1). Co-crystals of LEV with QUE and MYR, invariably obtained as non-stoichiometric EtOH solvates, i.e., LEV∙QUE∙xEtOH and LEV∙MYR∙xEtOH, respectively, had to be converted into their stable, unsolvated forms LEV∙QUE and LEV∙MYR with mild thermal treatment. LEV_2_∙HES and CIP∙QUE, on the contrary, were always obtained as unsolvated co-crystals. The four novel materials were then tested for their potential antimicrobial activity in comparison with the equivalent physical mixtures of APIs and co-formers.

Solid-state characterization for all the products was performed via powder X-ray diffraction (PXRD) and thermal gravimetric analysis (TGA) (see the Appendix A). Composition and purity of the products were confirmed using 1H NMR spectroscopy (see the Appendix A). 

## 2. Materials and Methods

All reagents and solvents used in this work were purchased from Sigma-Aldrich (Merck, Massachusetts, U.S.) or TCI Europe (TCI Europe N.V., Belgium -Tokyo Chemical Industry, Japan) and used without further purification.

### 2.1. Synthethic Methodologies 

#### 2.1.1. Mechanochemical Synthesis 

All the co-crystals were synthesized mechanochemically with liquid-assisted grinding (LAG) [53] with ethanol (100 μL); a Retsch MM200 Mixer Mill (Verder Scientific- Verder Group, Netherlands) was employed, operated for 2 h at a frequency of 25 Hz, with 5 mL agate jars and 2 agate balls (diameter 5 mm). A 1:1 stoichiometry of the reagents was first tried (0.5 mmoL, i.e., 169.12, 159.12, 75.57, 180.68, and 165.67 mg for QUE·2H_2_O, MYR, HES, LEV·0.5H_2_O, and CIP, respectively), which yielded the ethanol solvates LEV∙QUE∙xEtOH, LEV∙MYR∙xEtOH, and the unsolvated co-crystal CIP∙QUE. In the case of the 1:1 reaction of LEV∙0.5H_2_O with HES, the 2:1 co-crystal LEV_2_∙HES was invariably obtained together with excess HES. The stoichiometry of the reaction was thus modified into 2:1, yielding quantitatively more LEV_2_∙HES. All products were left to dry out at room temperature, collected from the jar, and analyzed with PXRD. No dependance on the amount of EtOH used in LAG was observed.

#### 2.1.2. Slurry in Ethanol 

The co-crystals LEV∙QUE∙xEtOH, LEV∙MYR∙xEtOH, and CIP∙QUE were also synthesized via slurry in ethanol (1 mL) of 1:1 stoichiometric mixture (2:1 in the case of the reaction between LEV and HES) of the reactants (same mmol and mg as in the mechanochemical synthesis). The suspensions were kept under stirring in the dark, to prevent a possible degradation of the flavonoids, and at room temperature for 3 days in a 10 mL glass vial closed with a PE pressure plug. The solid products were recovered and analyzed after filtration and drying. As the amount of ethanol detected in the two solids LEV∙QUE∙xEtOH and LEV∙MYR∙xEtOH was not constant, as observed in a number of syntheses, the co-crystals were subjected to mild thermal treatment (see below) that converted them to their stable, unsolvated forms, LEV∙QUE and LEV∙MYR, to be used for the antimicrobial tests.

#### 2.1.3. Crystallization from Solution 

Attempts at growing single crystals by synthesis or recrystallization of the solid products from ethanol either resulted in the formation of polycrystalline powders or, in the case of the solvated co-crystals, of poorly diffracting single crystals, due to rapid loss of ethanol even during short-time X-ray data acquisitions.

### 2.2. Solid-State Characterization

#### 2.2.1. Powder X-ray Diffraction 

Room-temperature powder X-ray diffraction patterns were collected using Bragg–Brentano geometry on a PANalytical X’Pert Pro (Malvern Panalytical-Spectris, Malvern, UK) automated diffractometer equipped with an X’Celerator detector (Malvern Panalytical-Spectris, Malvern, UK), using Cu Kα radiation (λ = 1.5418 Å) without a monochromator in the 3–40° 2θ range (continuous scan mode, step size: 0.033°; time/step: 30 s; Soller slit: 0.04 rad; anti-scatter slit: ½; divergence slit: ¼; 40 mA × 40 kV). 

#### 2.2.2. Variable-Temperature Powder X-ray Diffraction (VT-PXRD)

For all the co-crystals discussed in this work, powder X-ray diffractograms were collected in the 3−40° 2θ range, in open air using Bragg−Brentano geometry, with a PANalytical X’Pert PRO automated diffractometer, equipped with an X’Celerator detector, using Cu Kα radiation without a monochromator and an Anton Paar TTK 450 (Anton Paar, Graz, Austria) system for measurements at controlled temperature. 

#### 2.2.3. Thermogravimetric Analysis (TGA)

TGA measurements for all co-crystals were performed using a Perkin-Elmer TGA7 (Perkin-Elmer-Medtech, Shelton, CT, USA) instrument in the temperature range 30–300 °C under a N_2_ gas flow, at a heating rate of 10 °C min^−1^.

### 2.3. Solution ^1^H NMR Characterization

^1^H NMR spectroscopy was performed to ascertain stoichiometry and purity of the co-crystals. All the NMR spectra for starting materials and products discussed in this work were recorded with a Varian MR400 (Varian-Scientific Instruments, Palo Alto, CA, USA), operating at the frequency of 400 MHz on proton, equipped with PFG (Pulse Field Gradient) ATB (AutoSwitchable Broadband) Probes. 

### 2.4. Antimicrobial Activity Tests against Helicobacter pylori 

The co-crystals LEV∙QUE, LEV∙MYR, LEV_2_∙HES, and CIP∙QUE were subjected to antimicrobial testing against *H. pylori*. The antimicrobial activity was assessed using the broth microdilution method according to CLSI and EUCAST guidelines. All solutions/suspensions were tested in parallel, using *H. pylori* G27 and *H. pylori* 26695 strains with different standard methods to determine their minimal inhibitory concentrations (MICs) and minimal bactericidal concentrations (MBCs). Drugs, flavonoids, and co-crystals were tested in 11 progressive concentrations ranging from 512 to 0.5 µg/mL. Compound solutions/suspensions were all prepared following the same procedure: 20 mg of analyte in 10 mL of physiological solution (0.9% NaCl) to a final concentration of 2 mg/mL; physical mixtures were prepared simply by mixing drug dispersions/solution and flavonoid dispersions in stoichiometric ratios. The antibacterial assay was carried out using a 96-well microtiter plate; the first column of wells was filled with 100 μL of 2× Brucella broth supplemented with 10% fetal bovine serum (FBS), and the subsequent columns were filled with 100 μL of 1× Brucella broth supplemented with 5% FBS. Then, 100 μL of sample solution/dispersion were added in the first column, obtaining a concentration of 1 mg/mL. The two-fold serial dilutions were obtained by taking 100 μL from column 1 and mixing it with 1× broth in column 2 and then proceeding in the same way from column 2 to well 3, and so on. The volume withdrawn from the last column was discarded, leaving 100 μL in all the wells. Afterwards, 100 μL of bacteria diluted in the same supplemented broth medium were added to each well to a final concentration of 1.0 × 10^5^ CFU/mL. The whole procedure resulted in 1:2 serial dilutions ranging from 512 to 0.5 μg/mL from well 1 to well 11. Negative controls were used to verify that the compound solutions/suspensions were not contaminated, while a positive control was added to check for bacterial growth and fitness. The positive control was used as a comparison to evaluate the MIC and the MBC. Plates were incubated in a CO_2_-controlled incubator (9% CO_2_) at 37 °C and examined visually after 72 h. MIC values were defined as the lowest concentration of compound that inhibited the visible growth of bacteria after 72 h of incubation. For MBC determinations, 10 μL aliquots of diluted bacterial cultures around the MIC were spotted on Brucella broth agar supplemented with 5% FBS and incubated for 48 h in a 9% CO_2_ environment at 37 °C. MBC was defined as the lowest concentration of compound that prevented the growth of ≥99.9% of *H. pylori*. Each experiment was performed in triplicate to confirm the results.

## 3. Results and Discussion

### 3.1. Co-Crystallization of Levofloxacin with Quercetin, Myricetin, and Hesperetin 

As mentioned above, the co-crystallization of levofloxacin with quercetin or myricetin results in the 1:1 solvated co-crystals LEV∙QUE∙xEtOH and LEV∙MYR∙xEtOH. Recrystallization attempts always yielded tiny crystals of insufficient quality for single crystal X-ray diffraction; attempts at fast data collections failed, even at low temperature, due to the rapid loss of ethanol under the X-ray radiation. 

In order to obtain stable co-crystals LEV∙QUE and LEV∙MYR, suitable for antimicrobial tests, the solvated co-crystals were desolvated using mild thermal treatment. The desolvation process was followed by variable temperature powder X-ray diffraction (VT-PXRD). Modifications in the powder patterns of the two ethanolates can already be appreciated at 80 °C and at 30 °C for LEV∙QUE∙xEtOH and LEV∙MYR∙xEtOH, respectively. The heating process was stopped at 120 °C, which ensured complete desolvation of both co-crystals. 

Figure 1 and Figure 2 show a comparison of the powder patterns, all collected at room temperature, for reagents, solvated products, and unsolvated co-crystals resulting from the VT-PXRD process. It is interesting to observe that the two unsolvated co-crystals LEV∙QUE and LEV∙MYR appear to be isostructural.

At variance with the 1:1 stoichiometry observed in the case of LEV∙QUE and LEV∙MYR, the co-crystallization of levofloxacin with hesperetin invariably yielded a 2:1 stoichiometric product (LEV_2_∙HES). Figure 3 shows a comparison of the powder patterns for the two reagents and the co-crystal LEV_2_∙HES.

The comparisons in Figure 1, Figure 2 and Figure 3 allow us to see that mixing LEV with QUE, MYR, or HES leads to new compounds and not to physical mixtures of the starting materials, which would result in an overlay of the diffraction patterns of the reagents. 

TGA measurements, confirming the presence of solvent in the co-crystals of LEV with QUE and MYRz, as obtained from the synthesis, and the unsolvated nature of the LEV_2_∙HES co-crystal, and ^1^H NMR spectroscopic analyses that confirm chemical composition and stoichiometry for the LEV∙QUE, LEV∙MYR, and LEV_2_∙HES co-crystals used for the antimicrobial tests, can be found in the Appendix A, together with a comparison of the powder patterns for the products of the ball milling and slurry co-crystallization processes. 

### 3.2. Co-Crystallization of Ciprofloxacin with Quercetin

Co-crystallization of ciprofloxacin (CIP) with the three flavonoids was successful only in the case of quercetin, yielding the novel crystalline compound CIP∙QUE (See Figure 4. PXRD patterns). Due to the large difference in the solubility of CIP and QUE, the co-crystal could only be obtained via ball-milling and slurry, not from solution.

TGA measurement, confirming the unsolvated nature of the CIP∙QUE co-crystal, and ^1^H NMR spectroscopic analysis that confirms the co-crystal chemical composition and stoichiometry can be found in the Appendix A.

### 3.3. Antimicrobial Activity 

The MIC and MBC values for each entity considered in this work are reported in Table 1 and graphically represented in Figure 5. Under the experimental conditions used, the MIC values obtained for both LEV_2_∙HES and LEV∙QUE co-crystals and the antibiotic alone were identical for the *H. pylori* G27 strain. LEV_2_·HES and CIP∙QUE co-crystals exhibited equivalent antibacterial activity to their respective antibiotics alone against *H. pylori* 26695 strain. Nevertheless, there was a significant difference between the two data sets as the co-crystals contained significantly less of the antibiotic levofloxacin. Specifically, the co-crystal suspensions of levofloxacin and quercetin (LEV·QUE), levofloxacin and hesperetin (LEV_2_·HES), and ciprofloxacin and quercetin (CIP·QUE) contained, respectively, 50%, 33%, and 50% fewer antibiotics, in terms of moles of levofloxacin or ciprofloxacin. Remarkably, there was no significant difference between the co-crystal suspensions and the physical mixtures having the same relative molar amounts. These results may indicate that physical co-crystallization of the two chemicals occurred during the antimicrobial assays or that the antibiotics’ interaction mechanisms with the cellular membrane were altered due to the presence of flavonoids. On the other hand, the co-crystal and physical mixture containing myricetin (LEV·MYR, LEV + MYR) resulted in a lower antimicrobial effect in both strains, as seen in the higher MIC and MBC values. Notably, the flavonoids, namely, hesperetin, myricetin, and quercetin, demonstrated a considerably low antibacterial activity, incomparable to that of antibiotics or suspensions, with MIC values from 128 to 512 µg/mL.

Overall, we concluded that for *H. pylori*, the combination of flavonoids with an antibiotic capable of forming co-crystals may be a new and advanced approach to reduce the total amount of antibiotic used for treatment while maintaining the same antimicrobial efficacy. This result is in line with the recommendations and guidelines of the WHO [13] and the WGO [14] to reduce the use of antibiotics to prevent or at least slow the emergence and spread of antibiotic-resistant pathogenic strains.

## 4. Conclusions

In the need for alternatives to the common drug discovery process, to reduce costs, time, and energy investments, co-crystallization and supramolecular aggregation techniques are offering us a viable and eco-friendly route to design and prepare novel pharmaceutical materials with desired modified properties [19,54,55,56].

In this work, we have reported the preparation and characterization of a series of co-crystals obtained by co-crystallizing via solid-state solvent-free methodologies two antibiotics of the fluoroquinolone class, namely, levofloxacin and ciprofloxacin with the flavonoids quercetin, myricetin, and hesperetin. As a result, we prepared and characterized four novel co-crystals, namely, LEV∙QUE, LEV∙MYR, LEV_2_∙HES, and CIP∙QUE. LEV∙QUE and LEV∙MYR were obtained by desolvation of the corresponding xEtOH solvates. LEV∙QUE, LEV∙MYR, LEV_2_∙HES, and CIP∙QUE were then tested against two different strains of *H. pylori*, a bacterium included in the list of “priority pathogens”, according to the WHO and WGO [14,57]. Interestingly, the high MICs observed for the flavonoids against *H. pylori* G27 and 26695 strains under experimental conditions are in contrast with data presented in other studies [44,45]. Nonetheless, our approach led to an unexpected but worth reporting outcome. Indeed, while no significant difference in the antimicrobial activity was observed between the co-crystal suspensions and the physical mixtures used as comparison reference, the antimicrobial efficacy of the antibiotics was improved. Our research reveals that the combination of CIP and LEV with flavonoids, whether as co-crystals or as physical mixtures, yields equivalent results in MIC and MBC but permits the use of a reduced antibiotic dose (between half and two thirds in moles). Moreover, the fact that the co-crystal and the physical mixture perform equally well may indicate that, under the experimental conditions, either the antibiotics LEV and CIP form aggregates with flavonoids that have similar properties to the co-crystals or their presence enables a different mechanism of interaction with bacterial cell membranes, improving the overall efficacy.

The antimicrobial activity tests suggest that there may be an additive or synergistic effect between the antibiotics LEV and CIP, interfering with the bacterial DNA replication, and the flavonoids HES and QUE, targeting the *H. pylori* TR HP1043, whether as physical mixtures or co-crystals. Although the proposed multitarget approach may have major advantages in reducing the amount of antibiotics required, further investigation is needed to confirm this hypothesis.

## Data Availability

The data presented in this study are all available in the main text.

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
