# Peer review of "Levofloxacin and Ciprofloxacin Co-Crystals with Flavonoids: Solid-State Investigation for a Multitarget Strategy against Helicobacter pylori"

_pharmaceutics, 2024, doi:10.3390/pharmaceutics16020203_

Round 1
Reviewer 1 Report (Previous Reviewer 3)
Comments and Suggestions for Authors
The manuscript has been corrected according to Reviewers' suggestions and can be accepted to publication.
Reviewer 2 Report (Previous Reviewer 4)
Comments and Suggestions for Authors
I appreciate your time in revising the manuscript, and I think the changes made have improved the paper.
This manuscript is a resubmission of an earlier submission. The following is a list of the peer review reports and author responses from that submission.
Round 1
Reviewer 1 Report
Comments and Suggestions for Authors
The manuscript presents the synthesis, solid-state analysis, and antibacterial studies of co-crystals formed by fluoroquinolone antibiotics (Levofloxacin and Ciprofloxacin) and selected flavonoids. The aim was to check if the co-crystals could enhance the antimicrobial activity tests against Helicobacter pylori. The Introduction clearly describes the objectives. The experiments are described in detail and the obtained results are justified. However, the Authors do not attach the cif file nor the crystallographic data for the co-crystal for which they describe quite a detailed crystal structure. They mentioned that the quality of data is not good, but anyway, the results are described and they are a starting point for the discussion of the powder diffractograms obtained for the other co-crystals. Being aware that the crystal with the disordered solvent molecule can be problematic, I would insist on publishing the crystal data and sending the cif file to the CCDC with the annotation about its quality.
Further, there is some mess in the Figures numbering and their references in the text.
I also wonder, what are the hkl indices for the planes in the region of 25-27 2theta, corresponding to the stacking interactions in the first co-crystal and the pure antibiotics crystals? Do they match?
Summing up, I find this paper interesting and worth publishing in Pharmaceutics after minor corrections.
Reviewer 2 Report
Comments and Suggestions for Authors
The idea of the manuscript is interesting because the need for new antimicrobials is high. The authors have prepared several cocrystals using two fluoroquinolones, ciprofloxacin and levofloxacin and three flavonoids. They succeeded in preparing several cocrystals, but they have not demonstrated a better antimicrobial activity for antibiotic resistant bacterial strains.
Hence, the authors should demonstrate some benefits of using these cocrystals instead of the antibiotic alone. For instance, a broader spectrum of the co-crystals than of the fluoroquinolone, or reduced side-effects etc. otherwise the manuscript is more appropriate for a chemistry journal.
Reviewer 3 Report
Comments and Suggestions for Authors
The manuscript entitled "Levofloxacin and Ciprofloxacin co-crystals with flavonoids: solid-state investigation for a mullti-target strategy against Helicobacter pylori." by C. Fiore et al. presents preparation of the co-formers containing antibiotic (levoxacin or ciprofloxacin) and flavonoid (quercetin, myricetin or hespeterin) and their antibacterial activities against H. pylori G27 and H. pylori 26695.
The goal of the research was preparation of co-crystals. The obtained products appeared to be not suitable for single crystal X-ray diffraction measurement. The authors tried to perform X-ray diffraction measurement for LEVO-QUE-EtOH, however, the quality of the obtained data was so poor that it was not publishable as the authors claim. This means the data has no scientific value and cannot be taken into account in any discussion. Other co-formers were investigated by the use of powder X-ray diffraction. The presented data in this paper does not unambiguously prove that any co-crystals were obtained. It seems to be that the using the term "co-crystals" is not correct.
None of the mixtures and co-formers presented in the manuscript showed any synergistic effects of an improved antibacterial activity in comparison to the antibiotic alone. The presented results do not confirm the included in the abstract thesis that:
"The preliminary results indicate that the presence of the co-formers, whether in the co-crystal lattice or in the physical mixture appear to enhance the antimicrobial activity of the fluoroquinolones allowing to drastically reduce the amount of API required to obtain the same effect on H. pylori."
The manuscript is recommended for rejection.
Reviewer 4 Report
Comments and Suggestions for Authors
Page 3, line 110: “A single co-crystal product was obtained in the case of the co-crystallization of ciprofloxacin (CIP) with …”
The wording here isn’t very clear, I would advise to reword to make clear to the reader that single crystals were obtained only for the CIP-QUE cocrystal
Page 3, line 112: “To this end, co-crystals LEVO-QUE-EtOH and LEVO-MYR-EtOH were first subjected to thermal treatment to ensure that the product tested were completely ethanol free.”
Please can the authors state here the resulting solid form after desolvation, and refer to the relevant section/Figure in the text
Page 3, line 133: “…operated at a frequency of 25Hz for 2 hours…”
Can the authors provide comment on why this milling condition was chosen? I ask this as a 2 hour milling period for LAG cocrystal formation is quite a long time and the milling period applied is an important parameter e.g. there have been numerous reports showing that higher stoichiometric cocrystals form at short times (few minute) eventually leading to the 1:1 ratio for longer milling periods. A number of studied from Franciska Emmerling have commented on this.
Page 3, line 134: “…liquid assisted grinding (LAG) (100 μL).”
Can the authors please state the total mass of solid used in the experiments along with the ratio of LAG liquid volume:total mass. This has been shown to be an important parameter in mechanochemistry that plays an important role in cocrystal formation. Numerous groups have commented on this in the literature over the years.
Page 4, Table 2: In its current form this table is not of much use as it requires more context e.g. are these the masses of each component used in all milling experiments? Is this the total mass of each component used over the course of all the experiments? A better use of such a table would be to tabulate the masses and volumes of components used in experiments e.g. masses of solid components, volume of LAG solvent for each experiment. Also there would be no need for this in the main paper and I think would be better in the SI.
Page 4, Section 2.2.1: The authors comment here on the challenges in working with the disordered solvent in the obtained crystal structure. I am curious whether or not the authors had tried using SQUEEZE (available via Platon and Olex) to deal with the EtOH in the structure? This is a commonly applied technique in the MOF community when refining structure with a lot of disordered solvent and it may help improve the final outcome of the structure refinement.
Although the authors state that the structure is not publishable, I think it is required that the SCXRD data is made available to the readers/reviewers
Page 6, line 247: “Tiny crystals….”
Please be specific, what size were the crystals (this should be recorded with the single crystal data anyway) – please provide this information to the reader.
Page 6, Figure 2: The panels here (except the zoomed area) are too small for the figure and do not allow the reader to view the detail. The authors should increase the size of the figure (perhaps making it a large square 2x2 panelled figure rather than just a 1x4 panel). In the current figures the unit cell axes are difficult to see, can this be clearly displayed for the reader to allow straightforward comparison to the description in the text.
Page 7, Figure 3: This is a nice Figure; however, it is not clear what the stacked yellow and grey coloured things are to the right of the image – if these are LEVO and QUE, why are they coloured differently to the other molecules that are shown? Also they seem to be shown in some sort of space filled representation, the scale of which seems off in comparison to the capped sticks representation of the other molecules.
Page 7, Figure 4: This figure is too small and the detail of the PXRD plot cannot clearly be viewed (apart from the large peak at ~5 deg 2Theta). Can the Figure please be made larger, or at least use a different y scale, or zoom in, to allow the viewer to clearly view and assess the smaller peaks in the pattern.
Also, the authors highlight the peaks that they ascribe to the pi-pi interaction in the structure. For this reason, it would be useful for the authors to provide the hkl indices of these reflections (there is a structure for this form) to allow the reader to compare against the crystal structure. The authors do not provide a reason for this assignment and I suggest that they provide their justification for this in the text. If it is based on this value of 2 theta being around 3.2 angstrom d-spacing then this should be clearly stated in the text.
Page 8, Figure 6 caption: ” The circle in blue is evidencing the area in the pattern that corresponds to the pi-stacking interactions described above.”
There is no blue circle in this figure, but there is in the previous Figure 5 – can this be corrected, please?
Page 8, line 310: “By comparing the XRPD patterns we observed that both the solvate and the de-solvated phase in the case of LEVO with QUE and MYR are isomorphous.”
By my eye, there does appear to be some differences (peak positions and relative intensities) between the LEVO-MYR_EtOH and the LEVO-MYR VT patterns. There are also differences between the LEVO-QUE-EtOH and the LEVO-QUE VT patterns (peak positions and relative intensities).
A better comparison would be provided if the authors presented the PXRD collected from the recovered sample (e.g., the desolvated form back at room temperature) as this would allow for direct comparison of peak positions (no need to factor in thermal expansion of the unit cell and associated shift of peak positions).
Page 8, Figure 5: As with the previous figure, there are differences between the EtOH and desolvated patterns that indicates there is most likely a change in structure of the LEVO-QUE and LEVO-MYR component of the structure(s). The authors should make this clear to the readers for all of these data. If the authors have any other data or reasons to suggest that the EtOH has been removed with no change in the cocrystal component then this must be presented and the results discussed sufficiently to present this to the reader. At the moment, I don’t think the authors have provided the evidence required to make a convincing case that no change in structure has occurred.
Also, the MYR pattern in panel b is too small and should be enlarged in the y-direction to show the pattern clearly to readers.
Page 9, Figure 7: the pi stacking region again highlighted on the figure. I would again advise that the reasons for this assignment are provided to the reader.
Page 12, line 427: “… co-crystallizing via solid-state solvent-free methodologies…”
These solids were obtained via liquid assisted grinding – although using little solvent, it is not a solvent free process. This statement should be corrected.
ESI comments
NMR data. The peaks need to be assigned to each component e.g. which peaks in are attributed to the API and which to the coformer. It should also be shown the calculated ratio of each component as derived from the NMR data for each sample tested.
TGA data. There seems to be no analysis of the mass loss shown in the TGA data. The mass loss should be analysed to try and calculate/estimate the amount of EtOH lost from these samples i.e., is it a stoichiometric loss or not? This should be done for all relevant samples for which ethanol is assumed to be within the structure(s).
Comments on the Quality of English Languagenone
Reviewer 5 Report
Comments and Suggestions for Authors
The paper labelled « Levofloxacin and Ciprofloxacin co-crystals with flavonoids: solid-state investigation for a multi-target strategy against Helicobacter pylori» by Fiore and co-workers details investigation of crystals between fluoroquinolones and flavonoids both on solid-state and antimicrobial aspects. Data presented are globally clear and consistent and, regarding the objectives and policies of the journal, publication could be envisaged in Pharmaceutics. However, despite the quality of the present article, some points should be considered before publication:
- I am a little bit puzzled by the overall poor quality of thermal data (DSC and TGA). They were put in SI and are not clearly discussed (apart from desolvation of ethanol solvates) in the text. Particularly, DSC are not satisfying in terms of quality (with no straight baselines, poor choice of scaling leading to barely visible endotherms…). Since the authors highlighted an isomorphous co-crystal phase (LEVO-QUE and LEVO-MYR), it is necessary to discuss on their differences of thermal behaviour, especially melting points. For TGA, y-scale should be more consistently chosen, and derivative should be removed as it does not help the reader. My main opinion: i) either the authors decide to keep the thermal data and then the latter should be polished and better presented, and the consequences should be discussed in the main text or ii) the DSC data should be removed as they are not really discussed by the authors and only TGA measurements (with better presentation) should be kept.
- I do not see the added value of the crystallization from solution, since it did not work properly. The authors wrote that quality of SC-XRD data were not sufficient for publishing the structures, but the images presented in the articles comes from these data? Either structural resolution is qualitative enough to be published, or then it is just a proposition of structural model. If some pictures with interplanar distances precise down to 0.01 angstrom cand be presented, then the overall crystal structure should be too. It is also required to compare calculated and experimental XRD pattern to ensure the quality of structural model. Moreover, even though data are not “qualitative enough”, they should be provided at least in SI (crystal system, space group, lengths and angles of the lattice, and of course R factors…).
- I think that the settings written for XRPD are wrong: 3-40° with steps of 0.033° (thus 1121 steps) and step duration of 20 seconds (!!!!) will lead to measurements of 373 minutes: more than 6 hours per analysis. I doubt about it. Please check, I am quite confident that measurements were performed in continuous mode (then step data and duration supplied by PANalytical are just arbitrary and inconsistent) and that XRD duration should be closer to 10-15 minutes…
Apart from these main remarks, few typos and writing remarks:
- There are some repetitions in the main text that should be avoided. For example, the introduction could be shortened (lines 106-107 are just repeating lines 71-74), or the question of dark experiments in solution which are presented in lines 153 and repeated at lines 245.
- “1” in 1H NMR should be in superscript.
- Please precise the atmosphere of the DSC measurements (it was only mentioned for TGA).
- Line 284 (figure 4 caption): “pattern” instead of “patter”